# Demulsifier-Inspired Superhydrophilic/Underwater Superoleophobic Membrane Modified with Polyoxypropylene Polyoxyethylene Block Polymer for Enhanced Oil/Water Separation Properties

**DOI:** 10.3390/molecules28031282

**Published:** 2023-01-28

**Authors:** Mengmeng Zhang, Mingxia Wang, Junwei Chen, Linfang Dong, Yuqin Tian, Zhenyu Cui, Jianxin Li, Benqiao He, Feng Yan

**Affiliations:** 1State Key Laboratory of Separation Membranes and Membrane Processes, Tiangong University, Tianjin 300387, China; 2School of Chemistry, Tiangong University, Tianjin 300387, China; 3School of Materials Science and Engineering, Tiangong University, Tianjin 300387, China; 4Ningbo Fengcheng Advanced Energy Materials Research Institute Co., Ltd., Ningbo 315500, China

**Keywords:** Pluronic F127, PVDF ultrafiltration membrane, oil/water separation

## Abstract

Demulsifiers are considered the key materials for oil/water separation. Various works in recent years have shown that demulsifiers with polyoxypropylen epolyoxyethylene branched structures possess better demulsification effects. In this work, inspired by the chemical structure of demulsifiers, a novel superhydrophilic/underwater superoleophobic membrane modified with a polyoxypropylene polyoxyethylene block polymer was fabricated for enhanced separation of O/W emulsion. First, a typical polyoxypropylene polyoxyethylene triblock polymer (Pluronic F127) was grafted onto the poly styrene-maleic anhydride (SMA). Then, the Pluronic F127-grafted SMA (abbreviated as F127@SMA) was blended with polyvinylidene fluoride (PVDF) for the preparation of the F127@SMA/PVDF ultrafiltration membrane. The obtained F127@SMA/PVDF ultrafiltration membrane displayed superhydrophilic/underwater superoleophobic properties, with a water contact angle of 0° and an underwater oil contact angle (UOCA) higher than 150° for various oils. Moreover, it had excellent separation efficiency for SDS-stabilized emulsions, even when the oil being emulsified was crude oil. The oil removal efficiency was greater than 99.1%, and the flux was up to 272.4 L·m^−2^·h^−1^. Most importantly, the proposed F127@SMA/PVDF membrane also exhibited outstanding reusability and long-term stability. Its UOCA remained higher than 150° in harsh acidic, alkaline, and high-salt circumstances. Overall, the present work proposed an environmentally friendly and convenient approach for the development of practical oil/water separation membranes.

## 1. Introduction

With the increasing discharges from the petrochemical industry, domestic sewage, and industrial leaks, achieving effective oil/water separation appears to be a critical issue in the treatment of oil wastewater [1,2,3,4]. Traditional methods of treating oily wastewater such as coagulation [5], physical adsorption [6], and centrifugation [7,8] have a number of drawbacks including being prone to secondary pollutants, low oil/water separation efficiencies, as well as high energy costs [9,10,11]. In addition, conventional methods make it difficult to effectively treat oil/water emulsion, especially when the oil concentration is less than the value of 400 ppm and the oil droplet size is less than 20 μm [12,13,14]. Therefore, developing a low-cost and high-performance separation approach with environmentally friendly properties for oil/water emulsion is considered of utmost importance and faces numerous challenges.

In recent years, membrane separation has been extensively used in oil/water separation owing to the advantages of high efficiency, low cost, and simple operation [15,16,17]. Two types of membranes are generally employed in the scientific community, namely hydrophobic/oleophilic and hydrophilic/underwater oleophobic membranes [18,19]. Generally, the former one is usually modified with hydrophobic agents and is mainly used for the treatment of water-containing oils [20,21,22], while the latter one is typically modified with various hydrophilic substances, which can endow the membrane with hydrophilic substances and enhance the surface roughness. The membrane surface exhibits superoleophobic properties when wetted by water, and can effectively prevent contact with oil droplets. Thereby, efficient oil/water separation can be realized [23,24,25].

According to Wenzel’s model, the properties of hydrophilic/oleophobic are associated with the chemical composition as well as the surface microstructure [26,27]. In the last decade, blending, graft polymerization, and surface coating methods have been widely used to design superwetting membranes [28,29,30]. Teng et al. [31] embedded epoxied SiO_2_ nanoparticles and polyethyleneimine on the poly (vinylidene fluoride) (PVDF) membranes via a dip-coating process to form a hydrophilic layer, and the modified membranes retained over 98% of the oil during oil/water separation. Jin’s group prepared a new acid polyacrylic grafted PVDF membrane with low oil adhesion by salt-induced phase conversion method [32,33]. The micro-nanostructures on the membrane surface displayed ultra-high separation efficiency for different oil/water emulsions. However, most conventional hydrophilic modified membranes are mainly effective for separating oil/water mixtures or oil slicks in water, rather than oil/water emulsions. The membrane technology for the treatment of emulsified oil, particularly for systems such as oil recovery wastewater, still faces many challenges.

As far as the crude oil recovery industry is concerned, the oil recovery fluid is a mixture of oil/water (O/W) and water/oil (O/W) emulsions, and demulsifiers are regarded as the critical materials for demulsification and oil/water separation. Under this direction, inspired by the chemical demulsification mechanism of demulsifiers, our group previously grafted the hyperbranched polyether demulsifier onto the surface of the SMA/PVDF membranes through an in situ alcoholization reaction [34]. Interestingly, the produced demulsifier-functionalized membrane demonstrated excellent demulsification and oil/water separation performance in SDS-stabilized O/W emulsions. Nevertheless, this hyperbranched polyether demulsifier exhibited a complex structure and required high fabrication costs, which is not conducive to its application.

Demulsifiers are made of alcohols, polyamines, or phenolamine resins as starting agents, and polymerized with propylene oxide, ethylene oxide, etc. [35,36], and most of them usually have numerous polyoxypropylene polyoxyethylene block polymer branched chains. Inspired by this special chemical structure of the demulsifiers, a typical fragmentary structure of demulsifiers, namely polyoxypropylene polyoxyethylene block copolymers, was fixed to the PVDF membranes to provide a functional surface for improved separation of O/W emulsions. First, a typical polyoxypropylene polyoxyethylene block polymer (Pluronic F127) was grafted onto SMA by an esterification reaction. Then, the Pluronic F127-grafted SMA (abbreviated as F127@SMA) was blended with PVDF to prepare the F127@SMA/PVDF ultrafiltration membrane. The preparation procedure of the membrane is displayed in Figure 1. The chemical structure, surface morphology, antifouling performance, and wettability of the modified membrane were thoroughly investigated. Additionally, the separation effectiveness of the SDS-stabilized emulsions, the long-term stability, and the chemical durability were assessed. Interestingly, the as-prepared polyoxypropylene polyoxyethylene block polymer functionalized membranes demonstrated both extremely excellent oil/water separation properties and extremely low underwater oil-adhesion capability, opening a new route for the realistic surfactant-stabilized oil/water separation.

## 2. Results and Discussion

### 2.1. Characterization of F127@SMA

The FT−IR spectra of F127, SMA, and F127@SMA are presented in Figure 1a. In the spectrum of F127, the absorption peaks at 1100 cm^−1^ and 3400 cm^−1^ were ascribed to the vibrations of the C−O−C and −OH bonds, respectively. Moreover, the absorption peaks of the conjugated C=O bond in SMA occurred at 1780 cm^−1^ and 1850 cm^−1^, while these peaks disappeared in the spectrum of the F127@SMA. Furthermore, the characteristic peaks at 1100 cm^−1^ and 1720 cm^−1^ were depicted in the F127@SMA, which were attributed to the C−O−C and O=C−O vibrations, respectively. The FT−IR results demonstrated that the F127@SMA was successfully prepared.

The success of the grafting could be further verified by the analysis and comparison of the changes in the spectra of F127, SMA, and the F127@SMA by ^1^H NMR characterization, as shown in Figure 1b. From the spectrum of F127, it can be seen that the chemical shift at 3.5 ppm represented the −CH_2_− and −CH− groups on the backbone of polyether, and the peaks at 1.1 ppm represented the −CH_3_ group. In the spectrum of SMA, the chemical shift at 1.1–2.4 ppm was caused by the −CH_2_− and −CH− groups on the main chain, and the peaks at 6.6–7.3 ppm were caused by protons of benzene rings on the side chain of SMA. In the spectrum of the F127@SMA, both the peaks of the benzene ring on SMA and the peaks of −CH_2_− and −CH− groups on the backbone of polyether can be found. Specifically, the chemical shift of the −CH_2_− group linked to the ester group in F127 was found at 4.4 ppm, and the peak at a chemical shift of 2.3 ppm represented the −CH− group linked to the carbonyl group on the SMA backbone. The aforementioned ^1^H NMR results further demonstrated the successful grafting of the F127 with SMA.

The TGA curves of SMA, F127, and F127@SMA are presented in Figure 1c. As can be seen, SMA and F127 showed one–step weight loss within the temperature range of 250–420 °C and 340–410 °C, respectively. However, it was found that the weight loss in the F127@SMA consisted of two phases. A weight loss (6.56%) occurred before 180 °C in the F127@SMA, which was mainly attributed to the elimination of H_2_O. In addition, the main weight loss of the F127@SMA happened in the temperature range of 300–417 °C, which was between SMA and F127. These results further proved that the F127@SMA was successfully prepared. 

### 2.2. Chemical Structure Characterization of Membrane Surfaces

The ATR−FT−IR spectra of the membrane surface are indicated in Figure 2a. From the extracted outcomes, the typical absorption peaks of PVDF can be observed for all the membranes. Among them, the peaks that appeared at 875 cm^−1^ and 1402 cm^−1^ were the vibration absorption peak of −CH_2_−, while the absorption peak of −C−F− was generated at 1173 cm^−1^ [37]. Compared with the peak of the PVDF membrane (M-0), the characteristic peak of O=C−O appeared at 1720 cm^−1^ after adding the F127@SMA, whereas the intensity was slightly improved with the increase of the F127@SMA content. The above results clearly suggested that the F127@SMA had been blended successfully into the PVDF matrix.

The chemical elements on the membrane surface were investigated via XPS measurements, indicated in Figure 2b. In the XPS of the M-0 membrane, the peaks with binding energy values of 285.1 and 688.1 eV were generated by C1s and F1s, respectively, while for the F127@SMA/PVDF membranes, new peaks with a binding energy of 533.1 eV were observed. These new peaks were attributed to O1s from the F127@SMA polymers. To facilitate observation of the elemental changes on the membrane surface, the content of the elements and the ratio of O and F are presented in Table 1. From M-0 to M-7, with the increase of F127@SMA content, the O/F ratio gradually increased. It was noteworthy that M-7 exhibited the lowest content of the F element (13.8%) and the highest content of the O element (18.1%). From these results, it can be concluded that the F127@SMA was successfully blended into the PVDF matrix.

### 2.3. Morphology and Pore Structure of Membranes

The morphological structure of the membranes was observed by FE-SEM measurements, and the images are indicated in Figure 3. As seen in Figure 3a, M-0 exhibited smooth and dense surface morphologies, as well as a honeycomb cross-section. While after blending with the F127@SMA, the modified PVDF membrane exhibited an asymmetrical construction in combination with a thick cortex on top and a porous finger-shaped structure on the bottom, as can be observed clearly from Figure 3b,e. In addition, it was discovered that the pore size at the top and bottom of the membrane increased remarkably by increasing the F127@SMA content, suggesting that the membrane surface became loose, which was beneficial in improving its permeability. This result was ascribed to the excellent hydrophilicity of the F127@SMA. What is more, the addition of the F127@SMA had a significant effect on the cross-sectional architecture of the membrane. Interestingly, the addition of the F127@SMA polymer changed the cross-section of the membrane from a denser honeycomb-like pore shape to a finger-like pore shape. Moreover, by increasing the content of the F127@SMA polymer, the macro-cavities in the sublayer also increased gradually, whereas the membrane thickness increased accordingly, which was ascribed to the swelling effect of hydrophilic additive in the membrane matrix [38]. The above-mentioned changes in the microstructure would affect the permeability of the membranes (to be discussed later).

The porosity and the average pore size of the membrane were analyzed to explore the influence of F127@SMA content on the microstructure of the PVDF membrane. As displayed in Figure 4a, the addition of the F127@SMA polymer improved the porosity of the PVDF membranes and increased their average pore size. This was because the F127@SMA is a hydrophilic polymer with a comparatively high amount of hydrophilic polyoxyethylene groups. As a hydrophilic polymer, the F127@SMA accelerated the diffusion rate between DMAc (solvent) and the water (non-solvent), contributing to the formation of greater porosity and larger pores. Figure 4 indicates that the porosity of M-0 was less than 40%, whereas the porosity of M-1 was close to 65%. Correspondingly, the average pore size of M-0 was only about 18 nm, whereas that of M-1 was over 40 nm. As the content of F127@SMA increased, the pore size also increased gradually. The average pore size of M-7 was close to 60 nm, which was over three times that of the PVDF membrane. This trend of changes in porosity and average pore size was consistent with the results observed by FE-SEM images.

There was no doubt that the above-mentioned macro-cavity structure enhanced the permeate flux of the modified membrane (to be discussed below). However, the mechanical properties were also affected. As can be observed in Figure 4b, M-0 has the highest tensile strength (2.8 MPa) and elongation at break (75.0%). Additionally, both the tensile strength and elongation at break gradually decreased with the increase of the F127@SMA polymer content. The mechanical strength and tensile strength of M-7 were also only 1.0 MPa and 13.7%, respectively.

### 2.4. Permeability and Surface Wettability of Membranes

The permeability of the membrane was evaluated by water flux and BSA retention, as exhibited in Figure 5a. From the acquired outcomes, it was found that the water flux (8.6 L·m^−2^·h^−1^) of M-0 was the lowest. The fluxes increased obviously with increasing F127@SMA content, and the flux of M-7 reached 400.5 L·m^−2^·h^−1^. As illustrated in Figure 4a, the above results for water flux were consistent with the results for membrane pore structure. Generally, the permeability of membranes was closely related to hydrophilicity [39]. As can be seen in Figure 5a, the BSA rejection rate decreased from 99.8% to 88.8% by increasing the F127@SMA content from M-0 to M-7. This result was caused by the increased content of F127@SMA in the membrane, resulting in a larger pore size.

The surface wettability of the membranes was measured by contact angle measurements. Figure 5b illustrates the variation of the water contact angle (WCA) for different membrane surfaces. The WCA of the M-0 membrane was around 114.0°, whereas the water droplets could maintain their initial size after 60 s, which suggested the episteme of inherent hydrophobicity of the M-0. However, the WCA gradually decreased when the F127@SMA was added. The WCA of M-1 declined from 59.5° to 21.5° in 60 s, and the WCA of M-3, M-5, and M-7 all declined to 0° in less than 30 s, which confirmed that the addition of the F127@SMA in PVDF membranes shifts the membrane surface from highly hydrophobic to superhydrophilic. The significant decrease in WCA resulted from the addition of hydrophilic chemical components (such as −CH_2_CH_2_O−, −OH, and −COOH) to membrane pores and the membrane surface [40].

The variation of the underwater oil contact angle (UOCA) for various membrane surfaces is illustrated in Figure 6a. It was found that the UOCA of M-0 was about 82.0°, which indicated the inherent oleophilicity of M-0. However, the UOCA gradually increased when the F127@SMA was added. More specifically, the UOCA of M-1 increased to 137.0°, and the UOCA of M-3, M-5, and M-7 were 160.0°, 157.0°, and 158.5°, respectively. In addition, different oils were selected to perform UOCA tests on the M-0 and M-3. As depicted in Figure 6b, the M-0 membrane displayed conspicuous under oleophilicity. The UOCAs of the M-0 surface against petroleum ether, toluene, dichloroethane, kerosene, and crude oil were 81.0°, 78.5°, 80.5°, 87.5°, and 83.0°, respectively. However, the UOCA of M-3 increased towards 155.5°, 150.5°, 151.5°, 163.5°, and 161.0°, which confirmed the underwater superoleophobicity for M-3. From these results, it was demonstrated that by adding the F127@SMA polymer, the membrane surface could be converted from extremely lipophilic to underwater superoleophobic.

Underwater dynamic oil resistance tests were conducted to further verify the oil-repelling properties of the hydrophilic membrane. We used a crude oil droplet as a detection probe. It can be observed from Figure 6c that the oil droplet was in full contact with the M-3 surface, then the droplet was gradually lifted upwards to move the droplet away from the M-3 surface. Throughout the lifting procedure, the oil droplet maintained spherical at all times, while no significant deformation was observed. However, when the same operation was performed on the surface of M-0, the oil droplet was first adsorbed on the membrane surface and then gradually absorbed by the membrane. The above-mentioned results further demonstrate that the M-0 membrane had a high affinity for crude oil, while the M-3 membrane exhibited extremely low underwater oil-adhesion properties. As a consequence, oil droplets were easily detached from the membrane surface that had been wetted with water, offering a favorable precondition for oil/water emulsion separation.

### 2.5. Antifouling Performance

To further verify the advantages of the produced polyoxypropylene polyoxyethylene block polymer modified PVDF membranes, in terms of antifouling properties, recirculation filtration experiments were carried out with the BSA solution [41,42], as indicated in Figure 7a. M-0 showed poor resistance to BSA contamination, with low flux recovery after rinsing with DI water. In contrast, after being cleaned by DI water, all modified membranes exhibited good flux recovery performance. The flux recovery ratio (FRR), total fouling ratio ($R_{t}$), reversible fouling ratio ($R_{r}$), and irreversible fouling ratio ($R_{ir}$) were used to illustrate antifouling capability of the membrane, while the resulting parameters of FRR, $R_{r}$, $R_{ir}$, and $R_{t}$ are depicted in Table 2. Generally, membranes presented better antifouling performance when the values of FRR and $R_{r}$ were higher and the values of $R_{ir}$ and $R_{t}$ were lower. As can be found in Table 2, the values of FRR and $R_{r}$ for M-0 were only 14.0% and 4.1%, whereas the values of $R_{ir}$ and $R_{t}$ were as high as 81.4% and 85.5%, indicating the poor antifouling nature of M-0. In contrast, the values of FRR and $R_{r}$ increased, while the respective values of $R_{ir}$ and $R_{t}$ declined significantly for the F127@SMA modified PVDF membranes. It was also noticeable that the values of FRR (91.8%) and *R*_r_ (19.6%) for M-3 were significantly higher than the respective values of the other membranes, whereas the values of $R_{ir}$ and $R_{t}$ of M-3 were the lowest among the studied membranes. These results suggested that M-3 had the most excellent antifouling properties, which effectively protect the membrane surface from oil contact.

By considering the mechanical strength, permeability performance, superhydrophilic/underwater superoleophobic capability, and antifouling properties, the M-3 membrane was selected to conduct subsequent experiments.

### 2.6. Separation of the O/W Emulsion

#### 2.6.1. O/W Separation Performance

The separation properties of the modified membrane (M-3) were assessed via SDS-stabilized O/W emulsions prepared by various oils, including petroleum ether, toluene, dichloroethane, kerosene, and crude oil. Photographs and microscopy pictures of the pristine emulsion and separated filtrates are displayed in Figure 8. The existence of emulsified oil within the water caused the SDS-stabilized emulsion to be cloudy, and the size of the oil droplets was discovered to be 2–9 μm. After filtration through the M-3 membrane, the turbid emulsions became clarified. It was revealed by optical microscope images that there were hardly any visible oil droplets in the filtrates. This phenomenon indicated that the oil in the emulsion was successfully separated by the F127@SMA/PVDF membrane (M-3).

The separation efficiency of the membrane was quantitatively analyzed via a total organic carbon analyzer (TOC), and the results are listed in Table 3. From Table 3, it can be seen that the TOC value for the crude oil/water emulsion reached 1082.7 ppm, which significantly declined to 7.8 ppm after separation with the M-3 membrane, and the oil removal rate was 99.3%. Similarly, the TOC removal rate for various O/W emulsions was above 99.1% after being carried out with the M-3 membrane. These results suggested that the modified membrane exhibited excellent separation efficiency for various emulsions.

#### 2.6.2. Anti−Crude Oil Fouling and Operational Stability

The contaminants in oil recovery wastewater are mainly crude oil, so the resistance of the membrane to crude oil contamination is extremely important for its long−term stable operation. Under this perspective, in order to assess the resistance of M-3 to fouling by crude oil, recirculation filtration experiments of crude oil/water emulsion were performed, and the results are displayed in Figure 7b. As can be observed from Figure 7b, M-0 had a poor anti−crude oil fouling performance as it had a very low FRR value (12.8%) after rinsing with DI water. On the contrary, the M-3 membrane exhibited excellent anti-crude oil fouling performance. It almost recovered its original flux completely with every washing and kept a constant emulsion flux after the implementation of the three−cycle filtration. The FRR of M-3 was as high as 97.2%, which was much bigger than that of M-0. This was because the M-3 membrane had almost no adhesion to crude oil as suggested in Figure 6c. On top of that, the oil droplets could be easily washed away with water, allowing the membrane to be reused. The aforementioned results clearly suggested that the M-3 membrane exhibited high resistance to crude oil contamination, which was promising for practical applications.

To assess the advantages of M-3 membranes for long-term use, ten cycles of filtration were carried out using crude oil/water emulsions. Each cycle for emulsion separation lasted for 120 min, and the acquired results are shown in Figure 9. It was discovered that the M-3 membrane filtration produced a steady flux of 272.4 L·m^−2^·h^−1^ and an extremely high oil removal rate of over 99.3%, whereas the contact angle remained unchanged even after ten cycles. From these results, it was further proved that the superhydrophilic and underwater superoleophobicity of M-3 were not destroyed, suggesting that the F127@SMA/PVDF membrane exhibits excellent reusability and long−term usability for emulsion separation.

The separation performance of some recently reported superwetting membranes for oil-in-water emulsion was compared, and the results are shown in Table 4. We found that the prepared F127@SMA/PVDF membrane achieved excellent oil/water separation efficiency. In addition, most membranes were only used to separate O/W emulsions prepared from light oils, while the F127@SMA/PVDF membrane could be used to separate emulsions with various oils, including crude oil.

#### 2.6.3. Stability of the Modified Membrane

The stability of the membrane in harsh environments determines the service life of its practical application. The chemical stability of the F127@SMA/PVDF (M-3) membrane was investigated. To assess the acidic/alkaline resistance of the M-3 membrane, it was soaked within an aqueous solution of various pH values (3.0–12.0) for one week. The results of UOCA, separation efficiency, and flux of crude oil/water emulsion are shown in Figure 10a,b. It can be found from Figure 10a that corrosion under acidic or alkaline conditions had no significant impact on the underwater oleophobic performance of the M-3 membrane. The recorded UOCA values were all still greater than 150° after a week of corrosion under different pH conditions. Furthermore, as shown in Figure 10b, the flux stayed relatively constant in the range of 252.0–272.0 L·h^−1^·m^−2^, and the separation efficiencies were all above 97.0%, indicating outstanding separation properties for crude oil/water emulsion. It should also be noted that there was a slight decrease in the UOCA value, oil/water separation efficiency, and flux after a week of immersion under strong acid conditions, which might be ascribed to the partial loss of the F127@SMA due to ester decomposition under the harsh acidic condition. In addition, M-3 was also soaked in a high concentration salt solution for one week. As shown in Figure 10c, the UOCA was almost unchanged, indicating that the M-3 membrane had a strong salt tolerance. These results displayed that the membrane was capable of treating emulsion in weak acid, alkaline, and salt environments.

## 3. Experimental

### 3.1. Materials

Polyvinylidene fluoride (PVDF, MW~2.0 × 10^6^ g/mol) was acquired from Zhejiang Suxing Plastic Material Co., Ltd. (Zhejiang, China). Poly styrene-maleic anhydride (SMA, MW~6.8 × 10^4^ g/mol) was purchased from Shenzhen Hucheng Plastic Additives Chemical Co., Ltd. (Shenzhen, China). Pluronic F127 (MW~1.3 × 10^4^ g/mol) was supplied by Sigma-Aldrich Trading Co., Ltd. (Shanghai, China). Moreover, Albumin Bovine Serum (BSA, MW~6.8 × 10^4^ g/mol) was obtained from Sora Biotechnology Co., Ltd. (Beijing, China), while oils (kerosene, 1,2-dichloroethane, toluene, and petroleum ether) were supplied from Kermel Chemical Reagent Co., Ltd. (Tianjin, China). Crude oil was supplied by the Shengli Oil Field (Dongying, China). All other employed reagents were bought from Fengchuan Fine Chemical Reagent Co., Ltd. (Tianjin, China), and were utilized directly without further purification. Moreover, the crude oil used in the experiments was diluted to 50% concentration with kerosene.

### 3.2. Preparation of F127 Grafted SMA Ploymer

The direct blending of the polyoxypropylene polyoxyethylene block copolymer F127 into PVDF for membrane formation is not the best strategy because F127 has good water solubility and will be lost during use. To ensure the stable existence of the polyoxypropylene polyoxyethylene ether copolymer on the membrane, Pluronic F127 was grafted onto SMA polymer before the fabrication of the membrane.

The process of grafting F127 onto SMA was as follows: first, F127 (28.0 g) and p-Toluenesulfonic acid (12.0 g) were both dispersed in 128.0 g THF in a three-necked flask in a 60 °C water bath. Then, SMA (46.0 g) was dissolved in 100.0 g THF and was added drop wisely into the above reaction solution. Afterward, the reaction was continued for 9 h after the addition of the SMA solution. The NaOH solution (0.5 mol·L^−1^) was added next, and the pH of the mixture was adapted to 7. Interestingly, white flocculent precipitates were continuously produced during this process. The precipitates were filtered and washed repeatedly with ethanol. The F127 grafted SMA polymer (abbreviated as F127@SMA) was gained after drying in a vacuum at 60 °C.

The chemical structures of the F127@SMA were determined by carrying out Fourier transform infrared (FT-IR, Nicolet Nexus-670, Nicolet, Wisconsin, USA), as well as proton nuclear magnetic resonance (^1^H NMR, AVANCE AV 400 MHz, Bruker, Karlsruhe, Germany) measurements using d6-DMSO as solvent. The thermal stability of the polymers was established by conducting thermogravimetric analysis (TGA, TG 209 F3 Tarsus, NETZSCH, Bavaria, Germany) from room temperature to 800 °C at a rate of 10 °C/min in a nitrogen atmosphere.

### 3.3. Preparation of the Membrane

The membranes were prepared by the NIPS method. Initially, PVDF (12 *wt*%) and the F127@SMA (0 *wt*%, 1 *wt*%, 3 *wt*%, 5 *wt*%, and 7 *wt*%) were dissolved in DMAc and reacted in a water bath at 70℃ for 9 h to obtain a homogeneous solution. Afterward, the mixture was left at room temperature for 12 h to ensure that the bubbles were sufficiently removed. After that, the casting solution was poured onto a dry glass substrate, and a liquid membrane with 200 μm thickness was made by a scraper. Finally, the liquid membrane was soaked in water at a temperature of 35 °C for 10 min to obtain an ultrafiltration membrane. The obtained membranes were recorded as M-0, M-1, M-3, M-5, and M-7.

### 3.4. Characterization of Membranes

#### 3.4.1. Structure and Surface Features

The chemical structure and chemical elements of the membrane’s surface were analyzed by conducting attenuated total reflectance-Fourier transform infrared spectroscopy (ATR-FT-IR, Nicolet Nexus-670, Nicolet, Wisconsin, USA) and X-ray photoelectron (XPS, NEXSA, Thermo Fisher, Massachusetts, USA) measurements. The structural morphology of the membrane was observed by field emission scanning electron microscopy (FE-SEM, Hitachi S-4800, Hitachi, Tokyo, Japan). Moreover, a layer of gold/palladium alloy (Cressington 108 Auto sputter coater, Ted Pella, Inc., California, USA) was sputtered onto the dry membrane samples before testing.

The porosity (*ε*, %) was evaluated via the dry-wet method [43,47] as indicated in Equation (1):(1)$$\varepsilon (\%)=\frac{m_{w}-m_{d}}{{\rho A\delta }_{0}}\;\times \;100\%$$
where $m_{d}\;$ and $\;m_{w}\;$ are the dry and wet weights of membranes (g), respectively. $\delta_{0}$ (cm) and *A* (cm^2^) represent the thickness and effective area of the membrane, and ρ denotes the density of water (0.998 g·cm^−3^).

The mean pore size (*r_m_*, nm) was calculated by the use of the Guerout–Elford–Ferry (GEF) method [48], as indicated in Equation (2):(2)$$r_{m}=\sqrt{\frac{\left(2.9-1.75\varepsilon \right)8\eta lQ}{\varepsilon A\Delta P}}$$
where η (Pa·s) represents the viscosity of water, Q (m^3^·s^−1^) refers to the volume of permeate water per time, l (m) and A (cm^2^) stand for the thickness and effective area of the membrane, and ΔP (MPa) denotes the pressure.

#### 3.4.2. Mechanical Properties

The mechanical properties of the membrane were determined by conducting tensile strength and elongation at break measurements. More specifically, the membrane sample was first cut into a rectangle (30 mm long and 10 mm wide) and stretched at room temperature with a universal testing machine (ZwickRoell Z 0.5 kN, ZwickRoell, Ulm, Germany) at a speed of 10 mm/min until fracture.

#### 3.4.3. Permeability and Surface Wettability

The permeation performance of the membrane was assessed via cross-flow filtration, and the test pressure was 0.1 MPa. Permeation flux (J, L·m^−2^·h^−1^) and rejection (R, %) were calculated by Equations (3) and (4), respectively:(3)$$J=\frac{V}{A\Delta t}$$
(4)$$R=\left(1-\frac{C_{p}}{C_{f}}\right)\times 100\%$$
where *A* (m^2^) refers to the effective area of the membrane, *V* (L) represents the solution volume, and Δt (h) denotes the permeation time. The BSA concentrations in the permeate ($C_{p}$) and in the feed ($C_{f}$) were calculated by the absorbance at 278 nm using a UV-spectrophotometer (Evolution201, Thermo Fisher, MA, USA).

The water contact angle (WCA) and underwater oil contact angle (UOCA) were determined by a contact angle goniometer (DSA30E Krüss GmbH, Hamburg, Germany). Typically, 2.0 µL droplets of water or oil were dropped and laid on the surface of the membrane. Contact angle data and images were acquired at room temperature.

#### 3.4.4. Antifouling Performance

The antifouling performance was characterized by performing dynamic cycling experiments with BSA solution (1000 mg·L^−1^). The entire contamination process consists of three parts, each of which was operated as follows. The membrane was first pre-pressurized with DI water for 0.5 h (0.15 MPa), followed by testing the initial water flux at a pressure of 0.10 MPa for 1 h. Afterward, the membrane was contaminated with BSA solution instead of DI water for 1 h, and then the water flux was remeasured after rinsing the membrane with DI water for another 0.5 h [49,50,51]. The flux recovery ratio (FRR), total fouling ratio ($R_{t}$), reversible fouling ratio ($R_{r}$), and irreversible fouling ratio ($R_{ir}$) were calculated by Equtions (5)–(8):(5)$$FRR=\frac{J_{w3}}{J_{w0}}\times 100\%$$
(6)$$R_{r}=\frac{J_{w3}-J_{w1}}{J_{w0}}\times 100\%$$
(7)$$R_{ir}=\frac{J_{w0}-J_{w3}}{J_{w0}}\times 100\%$$
(8)$$R_{t}=\frac{J_{w0}-J_{w1}}{J_{w0}}\times 100\%$$
where $J_{w0}$ is the flux of pure water, $J_{w1}$ denotes the flux of emulsion, and $J_{w3}\;$ represents the water flux after the third cleaning (L·m^−2^·h^−1^).

### 3.5. O/W Emulsion Separation

The various oil/water emulsions were made by mixing 1.0 g of oil (petroleum ether, toluene, dichloroethane, kerosene, or crude oil), and 0.2 g of SDS with 1.0 L of DI water and under ultrasound for an hour at 25 °C. The emulsion separation performance of the membrane was carried out using cross-flow filtration (as shown in Figure 2) at a pressure of 0.1 MPa. Moreover, an optical microscope (OLYMPUS BX43, Olympus Corporation, Tokyo, Japan), total organic carbon (TOC) analyzer (GE Innovox, SIEVERS, Boulder, CO, USA), and a particle size analyzer (Zetasizer Nano ZS90, Malvern Instruments Ltd, Malvern, UK) were used to determine the optical photos, oil content, and oil droplet size of the emulsions, respectively. The TOC removal rate (*R_TOC_*, %) was obtained by Equation (9):(9)$$R_{TOC}=\left(1-\frac{TOC_{p}}{TOC_{f}}\right)\times 100\%$$
where $TOC_{p}$ and $TOC_{f}$ represent the TOC values of the permeate and the feed emulsion (ppm), respectively.

The oil recovery wastewater has a certain degree of acidity and high salt content. Therefore, it was necessary to deeply investigate the stability of the modified membrane in solutions with various pH values and different salt concentrations. First, the membranes were soaked in aqueous solutions of different pH values for a week. Then, the membranes were taken out after a week of immersion and cleaned with DI water to test the UOCA, flux, and oil/water separation properties. Furthermore, the membranes were immersed in different concentrations of salt solutions. After a week of soaking, the salt solutions were removed, and the membranes were cleaned with water before the contact angle was tested.

## 4. Conclusions

In this work, a typical polyoxypropylene polyoxyethylene block polymer, Pluronic F127, was grafted onto SMA and then blended with PVDF to prepare the F127@SMA/PVDF ultrafiltration membrane by the NIPS method. The prepared F127@SMA modified PVDF membrane exhibited superhydrophilicity (WCA of 0°) and underwater superoleophobicity (UOCA over 150°), as well as extremely low underwater oil-adhesion and high oil/water separation efficiency (>99.1%). In addition, the modified membrane was reusable and demonstrated long-term operational stability. After ten cyclic filtrations using the modified membrane, the oil rejection rate and UCOA were over 99.3% and 157°, respectively. Most importantly, the modified membrane remained stable after immersion within harsh acidic/alkaline and high concentrations of NaCl solutions. In general, the proposed environmentally friendly and simple strategy has significant potential for efficient oil separation from oil/water emulsions.

## Data Availability

Not applicable.

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
