# Peer review of "Demulsifier-Inspired Superhydrophilic/Underwater Superoleophobic Membrane Modified with Polyoxypropylene Polyoxyethylene Block Polymer for Enhanced Oil/Water Separation Properties"

_molecules, 2023, doi:10.3390/molecules28031282_

Round 1

Reviewer 1 Report

Recommendation: Minor revisions needed.

Comments:

The paper by Zhang et al. contributes to an analytical approach to assessing the for enhanced oil/water separation properties on the superoleophobic membrane modified with polypropylene polyoxy-3 ethylene block polymer. The article gives an interesting scientific perspective on assessing the oil/water separation properties. Some issues should be addressed prior to publication.

1.  Figure 1. a. Please label the most critical FTIR peak that you discussed in the context.

2.  Figure 1. b. What is the that we are looking at for the proton NMR? Please be specific in the discussion and show a higher-resolution NMR graph.

2.  Figure 1. c. Please label the onset temperature on the TGA.

3.  Table 1. Please double-check the format of this table.

4.  Figure 3. Please be specific on the legend to help the reader better understand the graph.

Reviewer 2 Report

Authors describe Demulsifier-inspired superhydrophilic/underwater superoleo-
phobic membrane modified with polyoxypropylene polyoxy-

ethylene block polymer for enhanced oil/water separation

properties. Overall study is well designed and interesting, however, several aspects needs improvement;
1. Some text in keywords is repeated.
2. language of the manuscript should be overall revised carefully.
3. Introduction should be revised. Some recently reported work is not cited (i.e. https://www.sciencedirect.com/science/article/abs/pii/S1383586622021839).
4. FTIR of F127 and SMA is same. Please explain.
5. A comparison with recent similar studies should be added in the form of a table.
